# Biocomposites of Epoxidized Natural Rubber/Poly(Lactic Acid) Modified with Natural Substances: Influence of Biomolecules on the Aging Properties (Part II)

**DOI:** 10.3390/polym13111677

**Published:** 2021-05-21

**Authors:** Anna Masek, Stefan Cichosz

**Affiliations:** Institute of Polymer and Dye Technology, Faculty of Chemistry, Lodz University of Technology, Stefanowskiego 12/16, 90-924 Lodz, Poland; stefan.cichosz@dokt.p.lodz.pl

**Keywords:** poly(lactic acid), epoxidized natural rubber, polymer blend, natural additives, antioxidant

## Abstract

The aim of this study is to present the possible influence of natural substances on the aging properties of epoxidized natural rubber (ENR) and poly(lactic acid) (PLA) eco-friendly elastic blends. Therefore, the ENR/PLA blends were filled with natural pro-health substances of potentially antioxidative behavior, namely, δ-tocopherol (vitamin E), curcumin, β-carotene and quercetin. In this way, the material biodeterioration potential was maintained and the material’s lifespan was prolonged while subjected to increased temperatures or high-energy UVA irradiation (340 nm). The investigation of the samples’ properties indicated that curcumin and quercetin are the most promising natural additives that may contribute to the delay of ENR/PLA degradation under the above-mentioned conditions. The efficiency of the proposed new natural anti-aging additives was proven with static mechanical analysis, color change investigation, as well as mass loss during a certain aging. The aging coefficient, which compares the mechanical properties before and after the aging process, indicated that the ENR/PLA performance after 200 h of accelerated aging might decrease only by approximately 30% with the blend loaded with quercetin. This finding paves new opportunities for bio-based and green anti-aging systems employed in polymer technology.

## 1. Introduction

Hybrid plastic materials are referred as blends of two types of polymers: bio- and petro-based. Due to substituting a part of the petro-based resin with polymers and additives from renewable resources, they are considered to be more sustainable in comparison with their pure synthetic analogues [1,2,3,4,5,6]. Hybrid plastic materials are being created for many reasons. Yet, some major factors might be highlighted: rising environmental awareness, trial of incorporation of less noxious nature-derived substances into the polymer technology and economic reasons (the need to balance the costs of biodegradable polymeric matrices by manufacturers) [7,8,9,10,11,12,13].

As it was mentioned in the previous part [14] of this research, Hamad et al. [15] have gathered a broad range of precise information concerning blends containing poly(lactic acid) (PLA), their properties and applications. An additional subsection of the mentioned review debated on the future perspectives. The authors underlined the importance of PLA-based materials in 3D-printing and the development of porous PLA materials for biomedical applications, but they also drew attention to the importance of the recyclability and reformability of the PLA-containing blends. The last problem was also underlined in the recently published work by La Mantia et al. [16], namely, the lack of sufficient scientific background regarding the recycling of biopolymer blends. Contrary to synthetic polymer blends, PLA-containing materials exhibit a higher degradation rate, which is obviously attributed to the poly(lactic acid) content [17,18,19].

Clearly, it is more desirable to create a material which fully undergoes biodeterioration. Yet, since PLA blending is a promising modification technique enabling us to overcome some noticeable drawbacks of neat poly(lactic acid) [20,21,22,23,24,25,26] and hybrid bioplastic material creation helps to balance the production costs, the stabilization of PLA-based blends requires significant attention. Fortunately, similarly to pure bio- and synthetic polymers, the recycling of their blends might be carried out via mechanical or chemical recycling methods [1,16].

Nevertheless, research published by La Mantia et al. [27] confirmed that the presence of even small amounts of PLA in the poly(ethylene terephthalate) (PET) waste can noticeably affect the rheological properties of a recycled material. Moreover, scientists proved that the mechanical properties of the reproduced PET product were significantly changed.

Another example may be another work of Hamad et al. [28], which describes the properties of the PLA and polystyrene (PS) blend (PLA/PS = 50/50 wt%). According to the presented results, multiple extrusion and injection of a PLA/PS blend resulted in a drop in the values of stress and strain at break even after two processing cycles. Yet, according to the information reported by the authors, Young’s modulus was not significantly affected at this time. Unfortunately, while four processing cycles were performed, the Young’s modulus showed a reduction by 26%. The most noticeable property degradation was observed regarding the strain at break with 73% loss and stress at break with 79% drop. The authors attributed this phenomenon to the reduction in the polymers’ molecular weight with the multiple processing cycles.

The examples given above reveal how important it is to stabilize the PLA-containing blends. Hamad et al. [15] claim that it is possible to introduce antioxidative agents through the creation of composites containing some nanoparticles, e.g., nanoclays, silver nanoparticles, metal oxides or functional biopolymers.

According to Li et al. [29,30], the stabilization of rubber-like materials may be obtained with the addition of sulfur. Nonetheless, this research is a proposal of an alternative way to achieve this goal with some eco-friendly plant-derived substances, e.g., vitamins, flavonoids and carotenes. According to the information gathered in the literature, these plant-derived substances may play the role of antioxidants and anti-aging factors [31,32,33]. Lately, it was found that, among others, eugenol [34], rosmarinic acid [35], phytic acid [36] or catechol [37] might be successfully applied in polymeric materials in order to prevent their properties’ loss during aging. Additionally, the significant aspect of the research performed by our group is the broadening of the knowledge on the antioxidative potential of natural substances [38,39,40,41,42,43].

Recently, our team has proven that apart from the above-mentioned plant-derived compounds, hesperidin, which can be found in various citric fruits, may play the role of the effective anti-aging factor in silica-filled ethylene-norbornene copolymer (EN) based materials. Not only did it prohibit the carbonyl groups’ formation during the performed 400-h-long weathering aging, but also hesperidin prevented the loss of the mechanical properties of the polymer composite during weathering (initial tensile strength of the blends at the level of 40 MPa; after 400 h of aging: approximately 10 MPa for EN + silica and 30 MPa while hesperidin added) [44].

Therefore, the aim of this study is to investigate the effect of different plant-derived substances, whose anti-oxidant properties have been primarily assessed by our team [45,46,47,48]. δ-tocopherol (vitamin E; δ-TF), curcumin (CM), β-carotene (β-CT) and quercetin (QU) were chosen regarding the elastic blends of epoxidized natural rubber (ENR) and poly(lactic acid) stabilization. According to the previous research, the above-mentioned biomolecules may play the role of effective natural antioxidants, simultaneously being relatively cheap [45,46,47,48].

Thanks to the incorporation of natural additives, material biodeterioration potential described in the previous part of the research [14] might be maintained and the composites’ lifespan prolonged while subjected to increased temperatures or UV irradiation. The stabilization effect was tested within the 200 h-long thermo-oxidative and UV accelerated aging processes. Thus, it was proven that some of the mentioned natural additives may successfully play the role of antioxidants in the investigated elastic and eco-friendly ENR/PLA blends.

## 2. Materials and Methods

### 2.1. Materials

Polymer matrixes employed in this research: epoxidized natural rubber (ENR) (Epoxyprene ENR-50; 50 mol% epoxidation) obtained from Kumpulan Guthrie Berhad (Kuala Lumpur, Malaysia) and poly(lactic acid) (PLA), Ingeo 4043D purchased from Nature Works (Minnetonka, MN, USA). Lauric acid (97% purity), 1,2-dimethylimidazole (highest available purity) provided by Sigma-Aldrich (Darmstadt, Germany) and elastin hydrolysate purchased from Proteina (Lodz, Poland) were used as a cross-linking system. Natural additives, namely, δ-tocopherol (δ-TF), CM—curcumin (CM), β-carotene (β-CT), quercetin (QU), of the highest available purities were purchased from Sigma-Aldrich (Darmstadt, Germany). Structures of natural additives are shown in Figure 1.

### 2.2. Preparation of ENR/PLA Samples

Poly(lactic acid) was conditioned for 24 h at 70 °C in a laboratory oven (Binder, Tuttlingen, Germany) before being incorporated into epoxidized natural rubber. All mixture components (Table 1) were mixed in a micromixer (Brabender Lab-Station from Plasti-Corder with Julabo cooling system) at 160 °C for 30 min (50 rpm). Next, such prepared material was put between two roll mills (100 mm × 200 mm)—roll’s temperature: 20–25 °C, friction: 1:1.1, time: 2 min. The last step was to form the plate-like samples between two steel molds (with the use of Teflon sheets separating the mixture from the mold) in a hydraulic press—temperature: 160 °C, time: 60 min, pressure: 125 bar. 

### 2.3. Accelerated Aging of the Materials

#### 2.3.1. Thermo-Oxidative Aging

Thermo-oxidative aging was performed in a laboratory oven (Binder, Tuttlingen, Germany) at 70 °C for 200 h. Four samples of each ENR/PLA blend were placed in an oven and were taken out individually after, respectively, 50 h, 100 h, 150 h and 200 h.

#### 2.3.2. UV Irradiation

ENR/PLA blend samples were mounted in the special holders in the Atlas UV 2000 apparatus (Duisburg, Germany). Again, specimens were taken out individually after, respectively, 50 h, 100 h, 150 h and 200 h. The aging cycle consisted of two alternating segments: day segment (240 min, 60 °C, UV irradiation: 0.7 W/m^2^) and night segment (120 min, 50 °C, no UV radiation).

### 2.4. Methods of Polymer Blend Sample Characterization

#### 2.4.1. Swelling in Toluene

The experiment aim was to measure the weight of the sample immersed in toluene before and after the swelling process. Thus, 4 samples of different shapes of each material were prepared (30–40 mg). The samples were weighed before the measurement (m_1_) and then put into the vessel with toluene. When 48 h were over, they were taken out and weighed in the swollen state (m_2_). The excess toluene was cleaned with the filter paper and each sample was immersed in ethyl ether for 1–2 s. Subsequently, the samples were dried to constant weight in an oven at 50 °C for 96 h and weighed when the time was over (m_3_). In order to analyze the potential cross-linking density, two parameters were calculated: m_rise_ = m_2_ − m_1_ (m_rise_ reveals how much the sample is swollen in the solvent), m_loss_ = m_3_ − m_1_ (m_loss_ informs about the low molecular weight compounds washed out during the swelling in toluene).

#### 2.4.2. Contact Angle Measurement

Contact angle measurements were carried out for: distilled water and 1,4-diiodomethane (droplet volume set for: 1 μL). Before the measurement, surfaces of polymer composite samples were cleaned with the use of acetone. OCA 15EC goniometer by DataPhysics Instruments GmbH (Filderstadt, Germany) equipped with single direct dosing system (0.01–1 mL B. Braun syringe, Hassen, Germany) was employed. Surface free energy is calculated with the Owens–Wendt–Rabel–Kaelble (OWRK) method.

#### 2.4.3. Tensile Tests

Mechanical properties, namely, tensile strength (TS) and elongation at break (Eb), were determined based on the ISO-37 with the use of a Zwick-Roell 1435 device (Ulm, Germany). Tests were carried out on a “dumbbell” shape (thickness: 1 mm, width of measured fragment: 4 mm, length of measured fragment: 25 mm, total length: 75 mm, width at the ends: 12.5 mm). The samples were stretched at a speed of 500 mm/min. On the basis of obtained results, the aging coefficient *K* was calculated according to the Equation (1) as a quotient of the product of *TS* and *Eb* after and before the performed aging process [49]:(1)$$K=\frac{{\left(TS\cdot Eb\right)}_{after\;ageing}}{{\left(TS\cdot Eb\right)}_{before\;ageing}}$$

#### 2.4.4. Color Change

Spectrophotometer UV-VIS CM-36001 from Konica Minolta. Sample color was described with the CIE-Lab system (*L*—lightness, *a*—red-green, *b*—yellow-blue) employed in this research. Then, color difference (Δ*E*), whiteness index (*W_i_*), chroma (*C_ab_*) and hue angle (*h_ab_*) values were calculated according to the equations given below (2–4):(2)$$\Delta E=\sqrt{{\left(\Delta a\right)}^{2}+{\left(\Delta b\right)}^{2}+{\left(\Delta L\right)}^{2}}$$
(3)$$W_{i}=100-\sqrt{a^{2}+b^{2}+{\left(100-L\right)}^{2}}$$
(4)$$C_{ab}=\sqrt{a^{2}+b^{2}}$$
(5)$$h_{ab}=\{\begin{array}{c} arctg\left(\frac{b}{a}\right),\;when\;a>0\wedge b>0\\ 180{}^{\circ}+arctg\left(\frac{b}{a}\right),\;when\;\left(a<0\wedge b>0\right)\vee (a<0\wedge b<0)\\ 360{}^{\circ}+arctg\left(\frac{b}{a}\right),\;when\;a>0\wedge b<0 \end{array}$$

#### 2.4.5. Mass Loss during the Degradation Process

In order to observe some mass changes, while samples were subjected to elevated temperature and UV irradiation, specially prepared samples were being weighed at the following times during the accelerated aging processes: 0 h, 50 h, 100 h, 150 h and 200 h.

#### 2.4.6. Scanning Electron Microscopy (SEM) Analysis

Scanning electron microscopy (Zeiss, ULTRA Plus, Oberchoken, Germany) was employed in order to examine the morphology of prepared ENR/PLA blend samples. Magnification was 250 and 1000 times.

## 3. Results and Discussion

The previous part [14] of this research study showed that it is possible to control the degradation rate of ENR/PLA blends with plant-derived fibers while subjected to increased temperature or UV irradiation, simultaneously causing their easier biodeterioration. Moreover, it was revealed that thanks to the organic–inorganic phase ratio, the control over the material degradation rate during thermo-oxidation and UV aging could be enabled. However, the incorporation of natural fibers into ENR/PLA blends without any additional anti-aging substances was the reason for their lower resistance to the two above-mentioned factors (temperature, UV), and only MMT was found to be able to prolong the lifespan of analyzed ENR/PLA blends.

Therefore, in this part of the research, different substances leading to the ENR/PLA blends’ stabilization are investigated. Some natural compounds of the potentially antioxidative properties [31,32,33] have been chosen: δ-tocopherol (δ-TF), curcumin (CM), β-carotene (β-CT), quercetin (QU). Their effect on the ENR/PLA blend properties before and after the accelerated aging processes is shown below.

Similarly to the previous part [14] of the study, analyzed polymer blends were investigated regarding the properties which may indicate some information about the degradation of the ENR/PLA blends during the accelerated aging processes. A swelling experiment was caried out in order to assess the cross-linking density as the crosslinks may stabilize the structure of the polymer network and prevent quick material degradation [50]. Moreover, the polymer composites exhibiting a higher polar component of surface free energy are known to be more prone to degradation during the aging processes regarding the ongoing radical reactions [51], e.g., oxidation. Thus, surface free energy was examined with the employment of contact angle technique. Finally, it was examined if the natural additives proposed in this research influence the mechanical performance of the analyzed ENR/PLA blends. Then, the effect of thermo-oxidative and UV accelerated aging was tested.

### 3.1. Characterization of Specimens before the Accelerated Aging Process

The swelling experiment, tensile tests and contact angle measurements were carried out in order to, respectively, assess the cross-linking density of the prepared material, investigate the influence of natural additives on the blends’ mechanical performance and predict the degradation potential of the prepared specimens. 

In Figure 2a, two parameters are presented: mass rise after the swelling process, which is proportional to the number of bonds created between different macromolecules, and mass loss during the swelling process, which is proportional to the amount of low molecular weight particles washed out in the swollen state [52]. Thus, it may be concluded that δ-TF and QU did not affect cross-linking density in a significant way as the samples are swollen similarly to the reference ENR/PLA blend. At the same time, in the case of the ENR/PLA samples with the addition of CM and β-CT, a great drop in mass rise after the swelling process was detected. This may indicate that these two compounds (CM, β-CT) led to the significant improvement in cross-linking density.

Furthermore, analyzing the data concerning the mass loss during the swelling process, it is visible that different amounts of low-molecular weight substances were washed out during the swelling of ENR/PLA. The reason for the observed differences could be the varied rate in the migration of the applied compounds to the surface of the polymer blend [52], e.g., the mass loss of the CM-loaded sample is the lowest, while for δ-TF addition the highest rinsing effect is detected.

Moving forward and analyzing the mechanical properties of prepared ENR/PLA blends (Figure 2b,c), some more variations between the samples are revealed. The specimen which was expected to have cross-linked the most and exhibited the lowest swelling degree, namely, ENR/PLA + β-CT, was revealed to have the lowest tensile strength and elongation at break. This could be explained with the fact that β-CT possesses in its structure many C=C bonds that can easily take part in cross-linking during the vulcanization process, therefore stiffening the material and preventing its elongation [53]. At the same time, the rest of the prepared samples revealed similar values of tensile strength on the level of the ENR/PLA reference blend or slightly lower. 

More variations can be found regarding the elongation at break (Figure 2c). Each natural additive, apart from β-CT, enabled obtaining the material which can elongate more than the reference ENR/PLA blend. The highest elongation is achieved in the case of the sample filled with δ-TF. This could be explained with the fact that its molecule possesses in its structure a long carbon chain that may work as a plasticizing agent [18].

The plasticization effect of the analyzed ENR/PLA blend specimens with an addition of δ-TF and CM may also be visible regarding the tensile stress values for the elongations of 100%, 200% and 300% (Table 2). In the case of the samples mentioned above, the tensile stress at any point is lower than for the reference ENR/PLA polymer blend. 

Moreover, according to the data gathered in Figure 2d, some variations regarding the surface free energy and both its components are revealed. Taking into consideration the presented results, specimens loaded with δ-TF and β-CT seem to be more hydrophilic. This is evidenced with contact angle measurements (Table 3) as these materials exhibit easier wetting with water (lower value of contact angle). On the other hand, QU incorporation into the ENR/PLA blend leads to the surface hydrophobization and the contact angle with water rises up to (101 ± 4)°, which is the highest of observed values.

Samples with the addition of δ-TF, β-CT and QU exhibit lower total surface free energy in comparison with the reference ENR/PLA blend. The only material containing natural additives and exhibiting surface properties similar to the reference sample is ENR/PLA + CM. The lowest value for the polar component of surface free energy is detected for the sample loaded with QU, which may indicate the highest stabilization potential regarding further aging tests (potentially less polar active centers prone to oxidation) [50].

Finally, the morphology of the prepared ENR/PLA blend samples was examined with the use of the SEM method. Images captured during the measurement are presented in Figure 3. The magnifications of 250 and 1000 times were applied.

Importantly, no signs of the blend components’ separation can be denoted. This means that the mixing process of ENR and PLA was sufficient and effective.

Moreover, performed morphology analysis proved that the natural additives employed in this study are fairly well dispersed in the polymer matrix and no big aggregates are present. Nonetheless, some inclusions might be denoted, e.g., specimens filled with β-CT and QU.

It is also visible that the blends’ surfaces are characterized by different regularity. The reference sample of ENR/PLA blend has an uneven and ragged surface, while specimens prepared with the addition of natural substances seem to be smoother. The only exception is the sample of QU-modified ENR/PLA blend.

### 3.2. Characterization of the Aging Impact

ENR/PLA blend samples were subjected to accelerated aging of two types: thermo-oxidative (increased temperature in the presence of air) and UV (chamber with UV lamps; 340 nm). During the aging processes, radical reactions might be initiated via temperature or UV irradiation and contribute to polymer chains’ crosslinking, oxidation or scission. This leads to some irreversible changes in the properties of the material subjected to the aging process [54].

The performed accelerated aging lasted for 200 h, and during the processes, after each 50 h, the tensile properties, mass of the polymer blend and color parameters were tested in order to assess the rate of the degradation process (Figure 4) and if the accelerated aging caused crosslinking resulting in the material stiffening or oxidation/scission that may be evidenced with, e.g., lowering mechanical performance of the investigated polymer blend [55,56].

Regarding the data gathered in Figure 4a, some differences between thermo-oxidative and UV aging could be found. Firstly, higher mass loss might be detected in the case of the first type of aging. It might be explained with the moisture content in the hydrophilic components of the loaded ENR/PLA blend [57,58], which is lost while the sample is subjected to the elevated temperature. 

For thermo-oxidative aging, the highest mass variations could be evidenced in the case of β-CT and δ-TF-loaded ENR/PLA blends. In turn, CM incorporation seems to cause a slower mass decrease. Yet, after 200 h of aging, the mass loss of the CM-filled specimen achieved a value higher than that observed for δ-TF addition. 

On the contrary, regarding the UV aging, weight loss occurs considerably slower and steadier. Moreover, the mass decrease values are the highest in the case of the reference ENR/PLA polymer blend, which means that the natural substances added to the composition could have possibly acted as successful stabilizers during the UV aging.

Moving forward to the graphs presented in Figure 4b and Figure 5, the detected color changes are quite significant for some polymer blend samples, and in some cases they reach the value of color change of approximately 20, e.g., δ-TF and CM-loaded specimens. Moreover, they become darker or lighter depending on the aging.

A color change higher than 3 means that variations evidenced for these samples could be easily visible with the human eye. Furthermore, it was observed that all substances steadily and constantly change their color upon the carried out aging processes (Figure 4b and Figure 6). Thus, the applied natural substances, apart from their application as natural antioxidants, might be very promising regarding the bio-based aging indicators for, e.g., food packaging [48,59].

It is worth noticing that samples of the reference ENR/PLA polymer blend after 100 h, 150 h, and 200 h of thermo-oxidative and UV aging were degraded noticeably. Thus, it was impossible to investigate the properties of the reference ENR/PLA polymer blend as it became too brittle to carry out the measurement.

Additionally, more data regarding the color change described with the parameters such as whiteness index, chroma and hue angle are presented in Figure 5. Based on the gathered information, some interesting conclusions might be made, e.g., the ENR/PLA + δ-TF specimen subjected to thermo-oxidation becomes darker and the color is more intense with the aging time. On the other hand, the color of the same sample subjected to UV irradiation becomes lighter. The color change representation is shown in Figure 6.

Moving forward and analyzing the mechanical properties of the prepared ENR/PLA blends during the aging process, it is visible that the reference material and the sample with the addition of β-CT are the most affected by the performed accelerated aging. Fortunately, δ-TF, CM and QU exhibit a stabilizing effect to a certain extent, depending on the used natural additive (Figure 4c–f). 

According to the gathered results, the most effective stabilization is visible for the QU-filled ENR/PLA blend. The sample’s performance slightly rose during the aging process and the elongation at break decreased the least in comparison with different samples.

The improvement in tensile strength upon the aging process observed for the QU-loaded sample might be explained with the previously evidenced ability of quercetin to cross-link ENR-containing blends [60]. On the other hand, the negative effect of β-CT on the ENR/PLA blend’s properties during the aging process might be explained with two factors: the above-mentioned ability to take part in the crosslinking process (and, thus, over-crosslinking of the polymer blend) and possible prooxidative behavior related to the not properly adjusted amount of β-CT in the polymeric mixture [61,62,63].

Considering Figure 4c,d, which refer to the aging performed in the elevated temperature, samples filled with δ-TF and CM become stiffer during the aging process as the tensile strength stays almost at the same level and elongation at break drops significantly.

This phenomenon may be explained with the ability of quercetin reported in the literature to crosslink the ENR [60]. It is possible that increased temperature or UV irradiation during the performed accelerated aging processes could have initiated the crosslinking of the polymer matrix and, thus, further stabilized the ENR/PLA blend. 

In turn, Figure 4e,f refer to the UV aging. It is visible that QU, again, exhibits a stabilizing effect and prevents the mechanical properties’ loss. However, CM also effectively delays the material degradation, leading to keeping the tensile strength at the same level during the aging process. 

A perfect confirmation for the stabilizing effect of these two compounds is presented in Figure 7. The values of the *K* coefficient, which compares the mechanical properties before and after the aging process, are presented for the samples prepared with and without potential natural anti-aging substances. Therefore, according to the data gathered in Figure 7, the positive effect of some among the applied compounds might be found (certain natural anti-aging substances are able to prevent a material’s significant performance drop during the aging processes). Additionally, information gathered in Table 4 helps to compare the results presented in this research with the previously carried out study, which debates on plant fibers’ incorporation into ENR/PLA blends.

It is clearly visible that QU is the substance of the highest stabilizing potential and it can be applied while the material is subjected to elevated temperature or UV irradiation. The maximum shift of the aging coefficient detected for the QU-loaded sample is from 1 to 0.6 ± 0.3 for 200 h-lasting UV aging and to 0.7 ± 0.3 for thermo-oxidative aging, which means a mechanical performance loss by, respectively, 40% and 30%. QU stabilizing activity may be explained as a synergic effect of antioxidative properties and the ability to crosslink ENR [47]. Moreover, according to the gathered data, CM might also be a quite efficient anti-aging factor for ENR/PLA blends while considering UV-resistance (the performance drop by approximately 50%).

### 3.3. Possible Stabilization Effect of the Natural Substances Employed in This Research

The aim of this subsection is to clarify the possible stabilization behavior of the natural substances employed in this research and provide the reader with some useful information regarding the polymer blends’ aging process analysis and the probable stabilization behavior of plant-derived substances employed in this research (Figure 8).

The overall mechanism of polymer material degradation might be divided into four stages, as follows: i.Initiation of the degradation reaction with external stimuli, e.g., temperature, irradiation, with simultaneous chemical bonds’ cleavage and formation of free alkyl radicals;ii.Initial propagation—reaction of alkyl radicals with oxygen and formation of peroxy/hydroxy radicals;iii.Further propagation—chain branching and transferring radical activity to another chain;iv.Termination and inactivated form creation with possible disproportionation reactions.

In general, polymer degradation leads to the shortening of the polymer chains’ length and, therefore, it provides a material with different properties [32,64].

Thermal degradation starts while the interatomic vibration energy is equal to or exceeds the energy of interatomic bond dissociation. Then, the bond cleavage occurs and two active macroradicals, which may further take part in the propagation processes, are formed. In turn, UV light leads to the polymer chain photodegradation. 

Similarly, it is initiated when the energy of absorbed radiation is greater than or at least equal to the dissociation energy of the individual bonds in the macromolecule. However, this type of degradation may also be initiated due to the presence of chromophore moieties which could shift their forms from basic singlet state to the excited singlet/triplet state, hence becoming incredibly reactive. In the case of PLA, the chromophore moieties are C=O carbonyl moieties [32].

On the other hand, regarding the structure of ENR, C=C bonds and unreacted oxirane rings are expected to be the most fragile to the free radicals’ presence, thus being prone to oxidation or further crosslinking during aging. Another problem could be the presence of some proteins which may affect the material’s resistance to elevated temperatures or UV irradiation [65,66]. 

Additionally, the above-described degradation processes are the perfect examples of avalanche reactions as the propagation step provides various different free radical forms, e.g., oxygen-incorporated (hydroxy/peroxy) and alkyl radicals. Thus, once initiated, the degradation process is highly challenging to stop.

The plant-derived substances employed in this research, namely, δ-tocopherol (δ-TF), curcumin (CM), β-carotene (β-CT), and quercetin (QU), exhibit strong antioxidant properties, hence they are promising natural antioxidants for polymer-based materials. According to various studies presented in the literature, they may easily react with free radicals generated during the degradation of the polymer matrix, acting as free radical scavengers and simultaneously being oxidized [64,67,68,69,70,71,72]. 

In this way, the chemical compounds that promote ENR/PLA degradation are eliminated and the deterioration of the polymer matrix is prevented or delayed (only the natural additive is degrading at that time and not the polymer matrix itself) [64,67,68,69]. Some possible stabilization mechanisms are presented in the works performed by Choe et al. [64] (δ-TF), Schneider et al. [67] (CM), Penicaud et al. [68] (β-CT), and Sokolova et al. [69] (QU). Some possible stabilization mechanisms, created on the basis of the available literature data presented in the works mentioned above, are given in Figure 8. 

## 4. Conclusions

Taking into consideration the gathered data, some natural substances potentially able to stabilize elastic blends of epoxidized natural rubber (ENR) with poly(lactic acid) (PLA) have been found. Nonetheless, further analysis is highly advised. Among the tested natural additives, quercetin was proven to be a substance of the highest stabilizing potential. Thus, it can be applied while the material is subjected to elevated temperature or UV irradiation. The maximum shift of the aging coefficient detected for the QU-loaded sample is from 1 to 0.6 ± 0.3 for 200 h-lasting UV aging and to 0.7 ± 0.3 for thermo-oxidative aging, which means a mechanical performance loss by only, respectively, 40% and 30%. Moreover, curcumin seems to also be a quite efficient anti-aging additive for ENR/PLA blends while considering UV-resistance (the performance drop by approximately 50%). Curcumin is also very promising as a future natural aging indicator. The hue angle of CM-filled samples might be shifted significantly from (70 ± 2)° to (58 ± 2)° for thermo-oxidative aging and up to (63 ± 2)° for UV aging. Additionally, it was evidenced that these two substances, namely, quercetin and curcumin, do not lead to a deterioration of the mechanical performance of the ENR/PLA blends while added to the polymer matrix. The presented research indicates that the anti-aging effect in PLA-containing blends could be obtained with certain plant-derived additives. This provides new opportunities for the creation of the materials characterized by facilitated recycling and controllable lifespan.

## Figures and Tables

**Figure 1 polymers-13-01677-f001:**
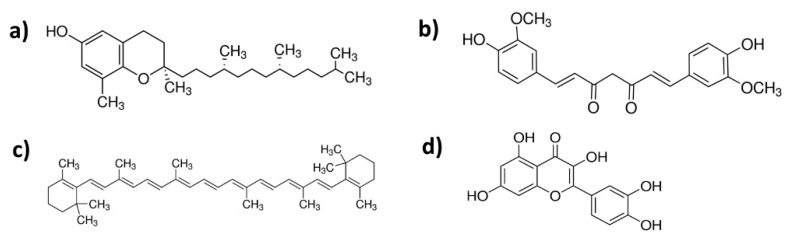
Structures of natural additives employed in this research: (**a**) δ-tocopherol (δ-TF), (**b**) curcumin (CM), (**c**) β-carotene (β-CT), (**d**) quercetin (QU).

**Figure 2 polymers-13-01677-f002:**
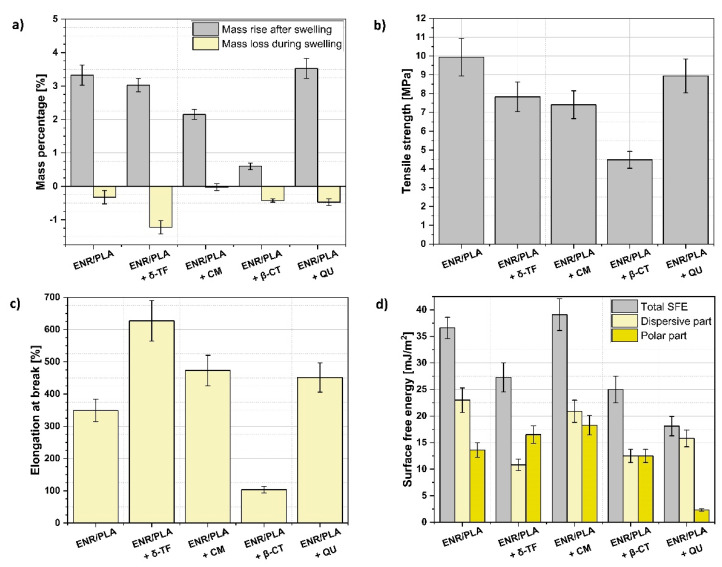
ENR/PLA blend properties before the accelerated aging processes: (**a**) analysis of the swelling experiment, (**b**) tensile strength of analyzed specimens, (**c**) elongation at break of investigated samples, (**d**) surface free energy and its components analysis.

**Figure 3 polymers-13-01677-f003:**
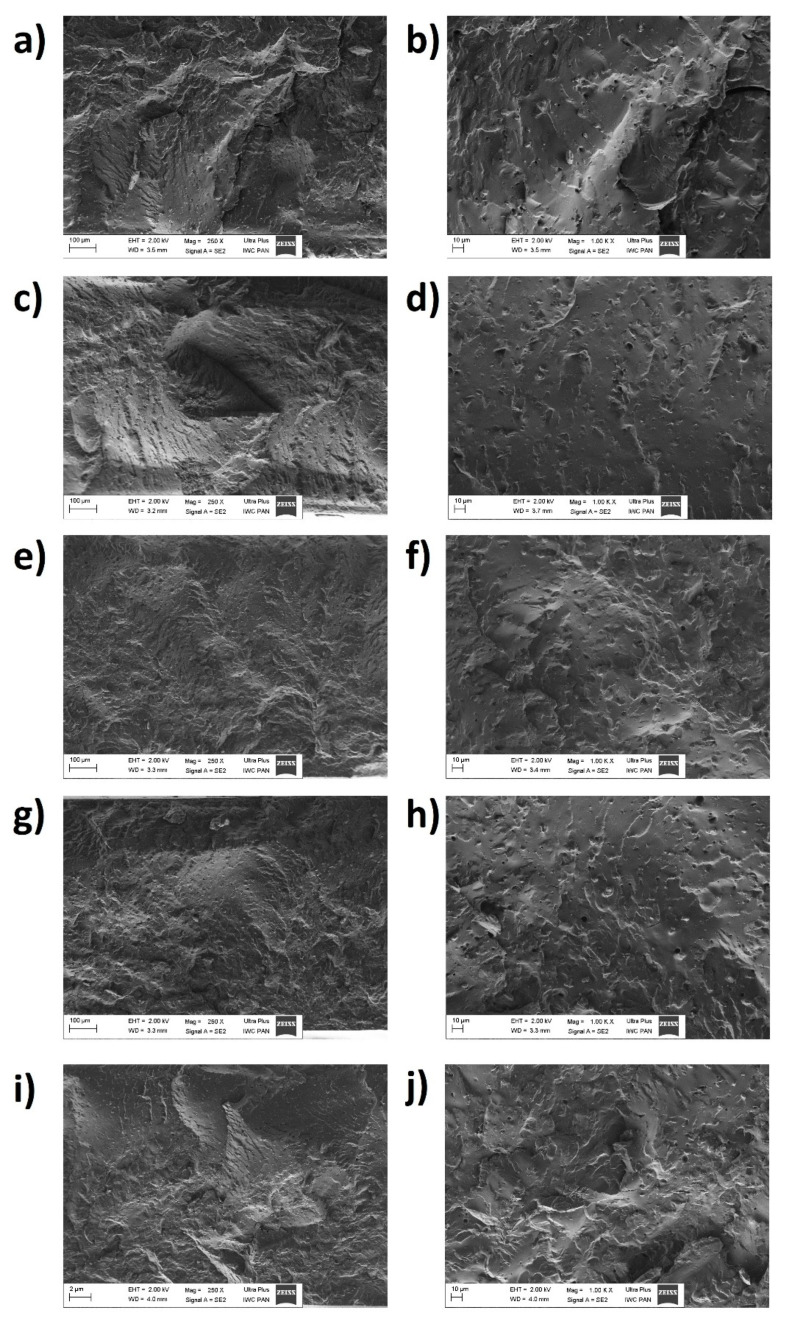
SEM images for: reference sample of ENR/PLA blend (**a**,**b**), ENR/PLA + δ-tocopherol (**c**,**d**), ENR/PLA + curcumin (**e**,**f**), ENR/PLA + β-carotene (**g**,**h**), ENR/PLA + quercetin (**i**,**j**) with magnifications, respectively, 250 and 1000.

**Figure 4 polymers-13-01677-f004:**
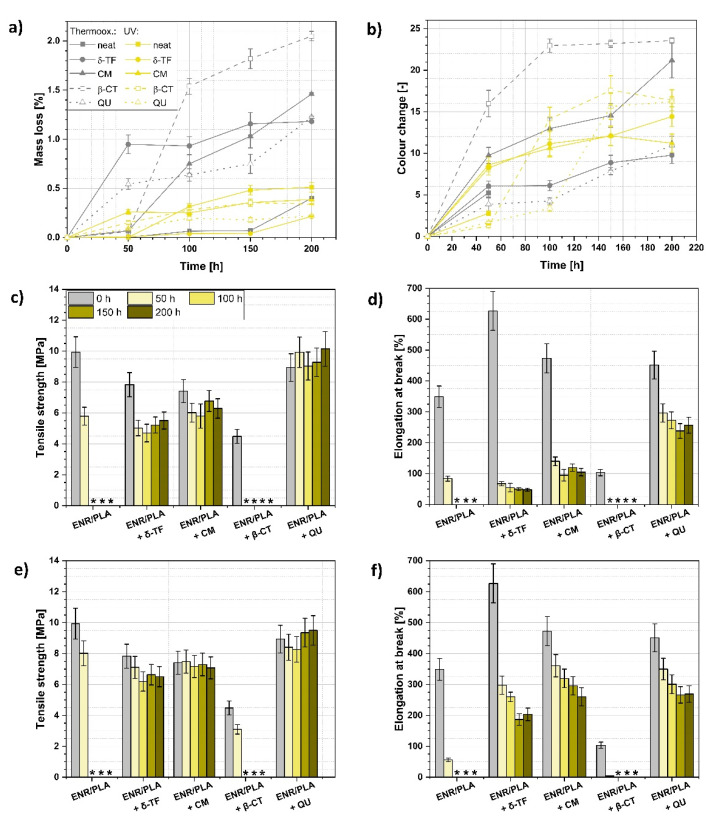
Investigation of the ENR/PLA blends’ properties before and after the accelerated aging processes: (**a**) mass loss during the thermo-oxidative and UV aging, (**b**) color change during the thermo-oxidative and UV aging, (**c**) tensile strength changes and (**d**) elongation at break variations during the thermo-oxidative aging, (**e**) tensile strength changes and (**f**) elongation at break variations during the UV aging. *—sample too brittle to be examined with the selected method; the measurement was impossible.

**Figure 5 polymers-13-01677-f005:**
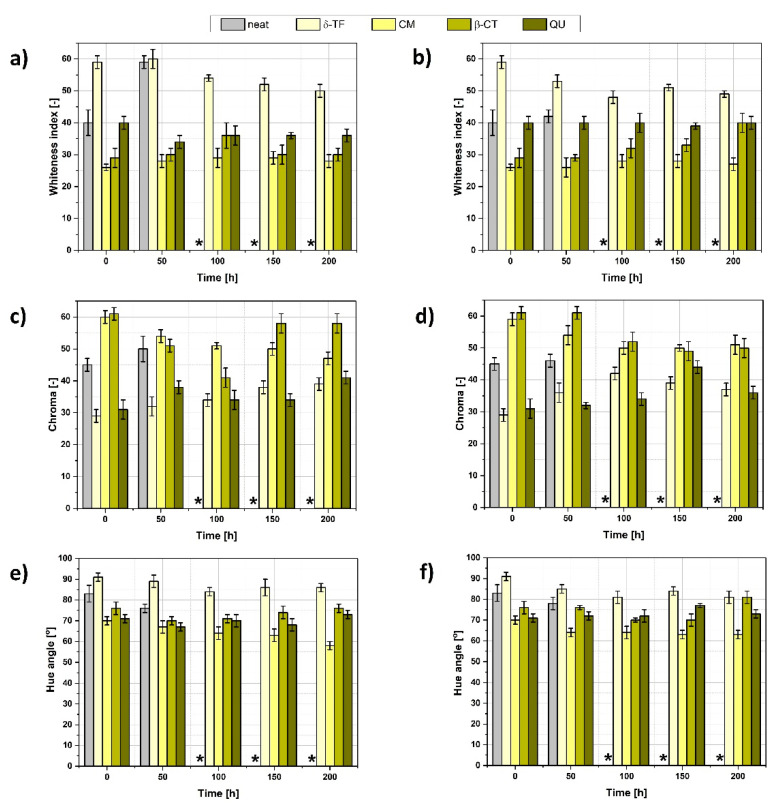
Additional parameters, namely, whiteness index, chroma, hue angle, contributing to color change of ENR/PLA blends and their variations during the thermo-oxidative (respectively: (**a**,**c**,**e**)) and UV (respectively: (**b**,**d**,**f**)) accelerated aging processes. *—sample too brittle to be examined with the selected method; the measurement was impossible.

**Figure 6 polymers-13-01677-f006:**
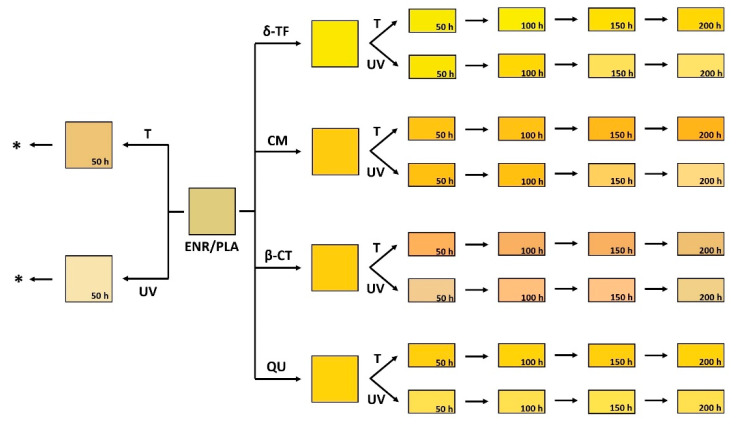
Color change representation for the prepared ENR/PLA blends before and after the accelerated thermo-oxidative (T), as well as UV-initiated (UV) aging processes. *—sample destroyed significantly.

**Figure 7 polymers-13-01677-f007:**
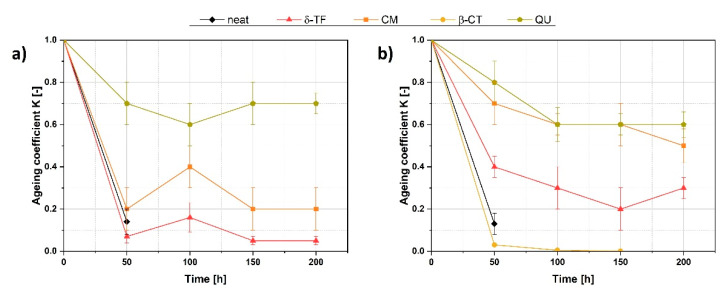
Aging coefficients attributed to the filled and unfilled ENR/PLA specimens at a certain aging time during thermo-oxidative (**a**) and UV (**b**) aging.

**Figure 8 polymers-13-01677-f008:**
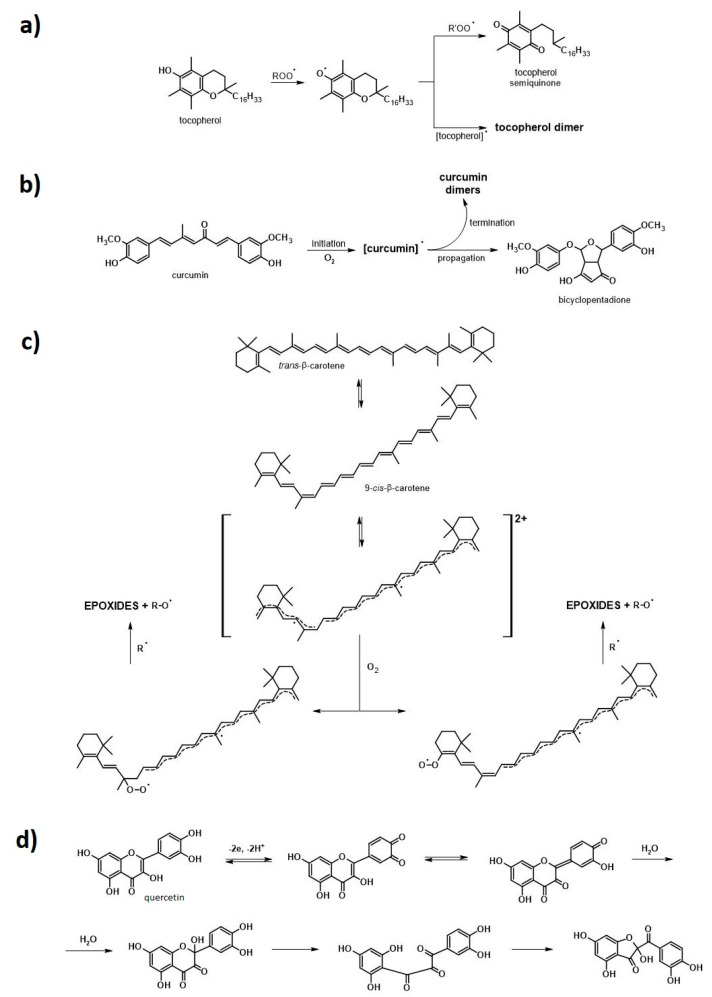
Possible mechanisms of free radical scavenging and oxidation behavior according to the available literature data: (**a**) δ-tocopherol (δ-TF), (**b**) curcumin (CM), (**c**) β-carotene (β-CT), (**d**) quercetin (QU).

**Table 1 polymers-13-01677-t001:** Composition of the polymer blend mixtures prepared for analysis in this research. Abbreviations: ENR—epoxidized natural rubber, PLA—poly(lactic acid), LA—lauric acid, DMI—1,2-dimethylimidazole, EH—elastin hydrolysate, δ-TF—δ-tocopherol, CM—curcumin, β-CT—β-carotene, QU—quercetin, phr—per hundred rubber (it means: for one hundred parts by weight of rubber there are x parts by weight of the substance).

Sample	Polymer Mixture Composition [phr]								
	ENR	PLA	LA	DMI	EH	δ-TF	CM	β-CT	QU
ENR/PLA	100	75	3	0.6	0.6	----	----	----	----
+ δ-tocopherol	100	75	3	0.6	0.6	3	----	----	----
+ curcumin	100	75	3	0.6	0.6	----	3	----	----
+ β-carotene	100	75	3	0.6	0.6	----	----	3	----
+ quercetin	100	75	3	0.6	0.6	----	----	----	3

**Table 2 polymers-13-01677-t002:** Tensile stress at elongation of 100%, 200%, 300% for the unfilled and filled ENR/PLA blend samples.

Sample	Tensile Stress [MPa] at Elongation of:		
	100%	200%	300%
ENR/PLA	4.5 ± 0.2	6.7 ± 0.3	8.8 ± 0.2
ENR/PLA + δ-tocopherol	3.3 ± 0.1	4.7 ± 0.2	5.8 ± 0.2
ENR/PLA + curcumin	3.8 ± 0.4	5.2 ± 0.5	6.3 ± 0.3
ENR/PLA + β-carotene	4.0 ± 0.3	-----	-----
ENR/PLA + quercetin	5.2 ± 0.3	6.8 ± 0.4	8.0 ± 0.5

**Table 3 polymers-13-01677-t003:** Water and diiodomethane contact angles for the unfilled and filled ENR/PLA blend samples.

Sample	Contact Angle [°]	
	Water	Diiodomethane
ENR/PLA	70 ± 3	57 ± 2
ENR/PLA + δ-tocopherol	67 ± 3	82 ± 1
ENR/PLA + curcumin	65 ± 2	60 ± 2
ENR/PLA + β-carotene	82 ± 1	80 ± 2
ENR/PLA + quercetin	101 ± 4	80 ± 3

**Table 4 polymers-13-01677-t004:** Comparison of aging coefficient values with the results presented in the previous part of the study [14]; CF—cellulose fibers, FF—flax fibers, MMT—montmorillonite. *—sample too brittle to be examined with the selected method; the measurement was impossible.

Sample ENR/PLA	Aging Coefficient *K* [-]—Thermo-Oxidation			
	50 h	100 h	150 h	200 h
+ CF [14]	*	*	*	*
+ FF [14]	0.004 ± 0.002	0.002 ± 0.001	0.002 ± 0.001	*
+ FF + MMT [14]	0.3 ± 0.1	0.3 ± 0.1	0.20 ± 0,08	0.20 ± 0,07
**Sample ENR/PLA**	**Aging Coefficient *K* [-]—UV Irradiation**			
	**50 h**	**100 h**	**150 h**	**200 h**
+ CF [14]	0.03 ± 0.01	0.06 ± 0.03	0.02 ± 0.01	*
+ FF [14]	0.02 ± 0.01	0.06 ± 0.02	0.06 ± 0.02	0.01 ± 0.01
+ FF + MMT [14]	0.2 ± 0.1	0.18 ± 0.07	0.15 ± 0.05	0.21 ± 0.08

## Data Availability

No data available.

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
