# Peer review of "Biocomposites of Epoxidized Natural Rubber/Poly(Lactic Acid) Modified with Natural Substances: Influence of Biomolecules on the Aging Properties (Part II)"

_polymers, 2021, doi:10.3390/polym13111677_

Round 1

Reviewer 1 Report

See the attachment.

Author Response

Institute of Polymer and Dye Technology

Technical University of Lodz

90-924 Lodz, ul Stefanowskiego 12/16, Poland

Tel.: +48 42 631 32 23, Fax: +48 42 636 25 43

May 17, 2021

Polymers

Dear Professor,

We are resubmitting our revised paper entitled Biocomposites of Epoxidized Natural Rubber/Poly(lactic acid) Modified with Natural Substances (Part II) by Anna Masek, Stefan Cichosz with a request to reconsider it for publication in Polymers.

We have carefully considered the Editor and Reviewers' comments. The manuscript was revised exactly according to these comments. The list of responses to the reviewer’s comments and corrections made in the manuscript is attached.

The manuscript has not been previously published, is not currently submitted for review to any other journal, and will not be submitted elsewhere before a decision is made by this journal.

For correspondence please use the following information:

corresponding author: Anna Masek

Institute of Polymer and Dye Technology

Technical University of Lodz

90-924 Lodz, ul Stefanowskiego 12/16, Poland

Tel.: +48 42 631 32 93

Fax: +48 42 636 25 43

Yours sincerely,

Ph. D., D.Sc. Anna Masek

All changes are marked with a green colour through whole manuscript.

Reviewer #1

This paper discusses the feasibility of utilizing the natural substances on the aging properties, and the useful contributions to this field will be expectable. However, there are some lines to be refined for more beneficial report.

The comments are listed below:

  1. The cause of selecting δ-TF, CM, β-CT and QU as plant-derived substance should be explained in the introduction. The role of them on the polymer matrixes is ambiguous.

Answer: We are grateful for this comment and believe that after this correction the plant-derived substance selection is clear. The description has been improved as follows: According to Lii et al. [29,30] stabilization of rubber-like materials may be obtained with addition of sulphur. Nonetheless, this research is a proposal of an alternative way to achieve this goal with some eco-friendly plant-derived substances, e.g., vitamins, flavonoids and carotenes. According to the information gathered in literature, these plant-derived substances may play a role of antioxidants and anti-aging factors [31–33]. Lately, it was found that among others eugenol [34], rosemaric acid [35], phytic acid [36] or catechol [37] might be successfully applied in polymeric mate-rials in order to prevent their properties loss during ageing. Additionally, the significant aspect of the research performed by our group is broadening of the knowledge on the antioxidative potential of natural substances [38–41].

Recently, our team has proven that apart from mentioned above plant-derived compounds, hesperidin, which can be found in various citric fruits, may play a role of the effective anti-ageing factor in silica-filled ethylene-norbornene copolymer (EN) based materials. Not only did it prohibit the carbonyl groups formation during the performed 400-hours-long weathering ageing, but also hesperidin prevented loss of the mechanical properties of the polymer composite during weathering (initial tensile strength of the blends at the level of 40 MPa; after 400 h of ageing: approximately 10 MPa for EN + silica and 30 MPa while hesperidin added) [42].

Therefore, the aim of this study is to investigate the effect of different plant-derived substances, which anti-oxidant properties has been primarily assessed by our team [43–46]. δ-tocopherol (vitamin E; δ-TF), curcumin (CM), β-carotene (β-CT) and quercetin (QU) were chosen regarding the elastic blends of epoxidized natural rubber (ENR) and poly(lactic acid) stabilization. According to the previous research, mentioned above biomolecules may play a role of effective natural antioxidants, simultaneously being relatively cheap [43–46].

  1. Tables 4 and 5 are unintelligible. Is graph better than table?

Answer: We agree with this comment and as Reviewer advised the mentioned tables have been changed into graphs.

  1. To be added more in detail in the conclusions might be better.

Answer: We tried to improve the Conclusions section and we hope that now it is clear for a reader and summarises the most important data presented in the manuscript: Taking into consideration gathered data, some natural substances potentially able to stabilize elastic blends of epoxidized natural rubber (ENR) with poly(lactic acid) (PLA) have been found. Nonetheless, further analysis is highly advised. Among tested natural additives quercetin was proven to be a substance of the highest stabilizing po-tential. Thus, it can be applied while the material is subjected to elevated temperature either UV irradiation. The maximum shift of the ageing coefficient detected for QU-loaded sample is from 1 to 0.6 ± 0.3 for 200 hours-lasting UV ageing and to 0.7 ± 0.3 for thermo-oxidative ageing which means the mechanical performance loss by on-ly, respectively, 40% and 30%. Moreover, curcumin seems to be also a quite efficient anti-ageing additive for ENR/PLA blends while considering UV-resistance (the per-formance drop by approximately 50%). Curcumin is also very promising as a future natural ageing indicator. Hue angle of CM-filled samples might be shifted significantly from (70 ± 2)° to (58 ± 2)° for thermo-oxidative ageing and up to (63 ± 2)° for UV age-ing. Additionally, it was evidenced that these two substances, namely, quercetin and curcumin, do not lead to a deterioration of the mechanical performance of the ENR/PLA blends while added to the polymer matrix. The presented research indicates that anti-ageing effect in PLA-containing blends could be obtained with certain plant-derived additives. This provides new opportunities for the creation of the mate-rials characterized by facilitated recycling and controllable lifespan.

Reviewer 2 Report

The subject of this research is fascinating. However, the manuscript needs some modifications as follows:

  • The English language should be improved throughout the manuscript.
  • 2.5.3. must be modified and explained well as follows:
    • According to which standard this test was carried out?
    • All specimen dimensions must be stated.
    • The loading rate must be stated.
    • Although Eq. 1 was mentioned in Ref. 42 and Ref. 34 in Ref. 42, it was suggested initially by 1016/j.crci.2011.11.013. Therefore, Ref. 42 must be replaced with the original one.
  • The term "Tensile tension" must be replaced by "Tensile stress".
  • Conclusions should be rewritten; it is not recommended to make a comparison with others in the Conclusion Section.

Author Response

Institute of Polymer and Dye Technology

Technical University of Lodz

90-924 Lodz, ul Stefanowskiego 12/16, Poland

Tel.: +48 42 631 32 23, Fax: +48 42 636 25 43

May 17, 2021

Polymers

Dear Professor,

We are resubmitting our revised paper entitled Biocomposites of Epoxidized Natural Rubber/Poly(lactic acid) Modified with Natural Substances (Part II) by Anna Masek, Stefan Cichosz with a request to reconsider it for publication in Polymers.

We have carefully considered the Editor and Reviewers' comments. The manuscript was revised exactly according to these comments. The list of responses to the reviewer’s comments and corrections made in the manuscript is attached.

The manuscript has not been previously published, is not currently submitted for review to any other journal, and will not be submitted elsewhere before a decision is made by this journal.

For correspondence please use the following information:

corresponding author: Anna Masek

Institute of Polymer and Dye Technology

Technical University of Lodz

90-924 Lodz, ul Stefanowskiego 12/16, Poland

Tel.: +48 42 631 32 93

Fax: +48 42 636 25 43

Yours sincerely,

Ph. D., D.Sc. Anna Masek

All changes are marked with a green colour through whole manuscript.

Reviewer #2

The subject of this research is fascinating. However, the manuscript needs some modifications.

The comments are listed below:

  1. The English language should be improved throughout the manuscript.

Answer: We are thankful for this comment. The whole manuscript has been checked once more considering the language and grammar.

  1. 2.5.3. must be modified and explained well as follows: According to which standard this test was carried out? All specimen dimensions must be stated. The loading rate must be stated.

Answer: Section 2.5.3. has been modified and all lacking information was added: Mechanical properties, namely, tensile strength (TS) and elongation at break (Eb), have been determined based on the ISO‐37 with the use of Zwick-Roell 1435 device (Ulm, Germany). Tests were carried out on a “dumbbell” shape (thickness: 1 mm, width of measured fragment: 4 mm, length of measured fragment: 25 mm, total length: 75 mm, width at the ends: 12.5 mm). The samples were stretched at a speed of 500 mm/min.

  1. Although Eq. 1 was mentioned in Ref. 42 and Ref. 34 in Ref. 42, it was suggested initially by 1016/j.crci.2011.11.013. Therefore, Ref. 42 must be replaced with the original one.

Answer: We are grateful for this comment. The reference has been changed.

  1. The term "Tensile tension" must be replaced by "Tensile stress".

Answer: We are sorry for this mistake. It has been corrected.

  1. Conclusions should be rewritten; it is not recommended to make a comparison with others in the Conclusion Section.

Answer: We tried to improve the Conclusions section and we hope that now it is clear for a reader and summarises the most important data presented in the manuscript: Taking into consideration gathered data, some natural substances potentially able to stabilize elastic blends of epoxidized natural rubber (ENR) with poly(lactic acid) (PLA) have been found. Nonetheless, further analysis is highly advised. Among tested natural additives quercetin was proven to be a substance of the highest stabilizing po-tential. Thus, it can be applied while the material is subjected to elevated temperature either UV irradiation. The maximum shift of the ageing coefficient detected for QU-loaded sample is from 1 to 0.6 ± 0.3 for 200 hours-lasting UV ageing and to 0.7 ± 0.3 for thermo-oxidative ageing which means the mechanical performance loss by on-ly, respectively, 40% and 30%. Moreover, curcumin seems to be also a quite efficient anti-ageing additive for ENR/PLA blends while considering UV-resistance (the per-formance drop by approximately 50%). Curcumin is also very promising as a future natural ageing indicator. Hue angle of CM-filled samples might be shifted significantly from (70 ± 2)° to (58 ± 2)° for thermo-oxidative ageing and up to (63 ± 2)° for UV age-ing. Additionally, it was evidenced that these two substances, namely, quercetin and curcumin, do not lead to a deterioration of the mechanical performance of the ENR/PLA blends while added to the polymer matrix. The presented research indicates that anti-ageing effect in PLA-containing blends could be obtained with certain plant-derived additives. This provides new opportunities for the creation of the mate-rials characterized by facilitated recycling and controllable lifespan.

Reviewer 3 Report

The authors have studied the influence of natural substances on the ageing properties of epoxidized natural rubber (ENR) and poly(lactic acid) (PLA) eco-friendly elas-10 tic blends. Their finding paves new opportunities for bio-based and green anti-ageing systems employed in polymer technology. To present a high-quality publication, follow revisions are advised:

  1. The language of English should be improved. There are many spelling and grammar mistakes through the manuscript.
  2. The numerical order for the subtitles are disordered: "1. Introduction; 2. Materials and Methods; 2. Results and discussion; 5. Conclusions. 
  3. Sulfur could be applied to influence the aging properties of rubber. This point should be mentioned in the introduction. Please refer to Energy Storage Mater. 34 (2021) 107-127, Energy Storage Mater. 27 (2020) 279-296. 

Author Response

Institute of Polymer and Dye Technology

Technical University of Lodz

90-924 Lodz, ul Stefanowskiego 12/16, Poland

Tel.: +48 42 631 32 23, Fax: +48 42 636 25 43

May 17, 2021

Polymers

Dear Professor,

We are resubmitting our revised paper entitled Biocomposites of Epoxidized Natural Rubber/Poly(lactic acid) Modified with Natural Substances (Part II) by Anna Masek, Stefan Cichosz with a request to reconsider it for publication in Polymers.

We have carefully considered the Editor and Reviewers' comments. The manuscript was revised exactly according to these comments. The list of responses to the reviewer’s comments and corrections made in the manuscript is attached.

The manuscript has not been previously published, is not currently submitted for review to any other journal, and will not be submitted elsewhere before a decision is made by this journal.

For correspondence please use the following information:

corresponding author: Anna Masek

Institute of Polymer and Dye Technology

Technical University of Lodz

90-924 Lodz, ul Stefanowskiego 12/16, Poland

Tel.: +48 42 631 32 93

Fax: +48 42 636 25 43

Yours sincerely,

Ph. D., D.Sc. Anna Masek

All changes are marked with a green colour through whole manuscript.

Reviewer #3

The authors have studied the influence of natural substances on the ageing properties of epoxidized natural rubber (ENR) and poly(lactic acid) (PLA) eco-friendly elastic blends. Their finding paves new opportunities for bio-based and green anti-ageing systems employed in polymer technology.

The comments are listed below:

  1. The language of English should be improved. There are many spelling and grammar mistakes through the manuscript.

Answer: We are thankful for this comment. The whole manuscript has been checked once more considering the language and grammar.

  1. The numerical order for the subtitles are disordered: "1. Introduction; 2. Materials and Methods; 2. Results and discussion; 5. Conclusions.

Answer: We are sorry for this mistake. It has been corrected.

  1. Sulfur could be applied to influence the aging properties of rubber. This point should be mentioned in the introduction. Please refer to Energy Storage Mater. 34 (2021) 107-127, Energy Storage Mater. 27 (2020) 279-296.

Answer: We are grateful for this valuable comment which may enrich our article. This point was mentioned in the introduction: Examples given above reveal how important it is to stabilize the PLA-containing blends. Hamad et al. [15] claim that it is possible to introduce antioxidative agents through the creation of composites containing some nanoparticles, e.g., nanoclays, sil-ver nanoparticles, metal oxides or functional biopolymers. According to Li et al. [29,30] stabilization of rubber-like materials may be ob-tained with addition of sulphur.

  1. Li, S.; Leng, D.; Li, W.; Qie, L.; Dong, Z.; Cheng, Z.; Fan, Z. Recent progress in developing Li2S cathodes for Li–S batteries. Energy Storage Mater. 2020, 27, 279–296.
  2. Li, S.; Fan, Z. Encapsulation methods of sulfur particles for lithium-sulfur batteries: A review. Energy Storage Mater. 2021, 34, 107–127.